# Unique Evolution of SARS-CoV-2 in the Second Large Cruise Ship Cluster in Japan

**DOI:** 10.3390/microorganisms10010099

**Published:** 2022-01-04

**Authors:** Haruka Abe, Yuri Ushijima, Murasaki Amano, Yasuteru Sakurai, Rokusuke Yoshikawa, Takaaki Kinoshita, Yohei Kurosaki, Katsunori Yanagihara, Koichi Izumikawa, Kouichi Morita, Shigeru Kohno, Jiro Yasuda

**Affiliations:** 1Department of Emerging Infectious Diseases, Institute of Tropical Medicine, Nagasaki University, Nagasaki 852-8523, Japan; abeh@nagasaki-u.ac.jp (H.A.); ushijima-yu@nagasaki-u.ac.jp (Y.U.); bb55420102@ms.nagasaki-u.ac.jp (M.A.); ysakurai@nagasaki-u.ac.jp (Y.S.); r-yoshikawa@nagasaki-u.ac.jp (R.Y.); takaaki@nagasaki-u.ac.jp (T.K.); ykuro@nagasaki-u.ac.jp (Y.K.); 2Graduate School of Biomedical Sciences, Nagasaki University, Nagasaki 852-8523, Japan; k-yanagi@nagasaki-u.ac.jp (K.Y.); koizumik@nagasaki-u.ac.jp (K.I.); moritak@nagasaki-u.ac.jp (K.M.); 3National Research Center for the Control and Prevention of Infectious Diseases (CCPID), Nagasaki University, Nagasaki 852-8523, Japan; 4Department of Laboratory Medicine, Nagasaki University Hospital, Nagasaki 852-8501, Japan; 5Infection Control and Education Center, Nagasaki University Hospital, Nagasaki 852-8501, Japan; 6Department of Virology, Institute of Tropical Medicine, Nagasaki University, Nagasaki 852-8523, Japan; 7Nagasaki University, Nagasaki 852-8521, Japan; president_kohno@ml.nagasaki-u.ac.jp

**Keywords:** SARS-CoV-2, evolution, phylogeny, cruise ship, Costa Atlantica, Japan

## Abstract

In the initial phase of the novel coronavirus disease (COVID-19) pandemic, a large-scale cluster on the cruise ship Diamond Princess (DP) emerged in Japan. Genetic analysis of the DP strains has provided important information for elucidating the possible transmission process of severe acute respiratory syndrome coronavirus 2 (SARS-CoV-2) on a cruise ship. However, genome-based analyses of SARS-CoV-2 detected in large-scale cruise ship clusters other than the DP cluster have rarely been reported. In the present study, whole-genome sequences of 94 SARS-CoV-2 strains detected in the second large cruise ship cluster, which emerged on the Costa Atlantica (CA) in Japan, were characterized to understand the evolution of the virus in a crowded and confined place. Phylogenetic and haplotype network analysis indicated that the CA strains were derived from a common ancestral strain introduced on the CA cruise ship and spread in a superspreading event-like manner, resulting in several mutations that might have affected viral characteristics, including the P681H substitution in the spike protein. Moreover, there were significant genetic distances between CA strains and other strains isolated in different environments, such as cities under lockdown. These results provide new insights into the unique evolution patterns of SARS-CoV-2 in the CA cruise ship cluster.

## 1. Introduction

Understanding the evolutionary patterns of severe acute respiratory syndrome coronavirus 2 (SARS-CoV-2) during the coronavirus disease (COVID-19) pandemic to identify significant mutation sites directly affecting viral characteristics is essential for establishing appropriate public health responses and developing effective countermeasures [1,2]. As of 1 October 2021, more than 230 million cases of SARS-CoV-2 infections have been confirmed worldwide, with 4.8 million deaths [3]. Due to the proofreading function of the viral proteins, SARS-CoV-2 isolates display relatively small phylogenetic distances among each other in the phylogenetic tree, making it difficult to clarify the genetic relationships among strains [4]. However, approximately four million complete genomes of SARS-CoV-2 have been deposited in the Global Initiative on Sharing All Influenza Data (GISAID) database (https://www.gisaid.org/ (accessed on 1 October 2021)). This extensive molecular surveillance of the virus has enabled researchers to identify critical mutations that might contribute to higher transmissibility or immune escape, such as D614G, N501Y, and E484K amino acid substitutions in the spike (S) protein [5,6,7,8].

Molecular surveillance studies of SARS-CoV-2 have been conducted in a number of countries or metropolitan cities to reveal evolutionary patterns in specific regions and environments [9,10,11,12]. These reports have successfully shown detailed trends of SARS-CoV-2 transmission and evolution in large populations or communities. Large-scale clusters on cruise ships, however, have rarely been reported, possibly because almost all cruise tours were canceled because of the high risks of easy transmission of the virus. The first and most studied cruise ship that experienced a large-scale cluster was the Diamond Princess (DP). A number of studies have been conducted on the epidemiology of this cluster in Japan, providing remarkable findings to understand the transmission process of the virus in the early phase of the COVID-19 pandemic [13,14,15,16]. Thus, the cruise ship cluster is informative for understanding SARS-CoV-2 infections in confined and mass-gathering places. One research group conducted genetic analysis using genome sequences of DP strains and revealed a unique evolution process of SARS-CoV-2 [17]. However, genetic studies of large-scale cruise ship clusters have rarely been conducted, probably because of the difficulty in obtaining samples, and it remains unclear how SARS-CoV-2 acquires diversity in a highly crowded and confined place.

The second large cruise ship cluster emerged in Japan in April 2020 on the Costa Atlantica (CA) cruise ship, which had been in repair in Nagasaki, Japan, since January 2020 [18]. There were only crew members on board, and COVID-19 diagnostic tests were performed on all 623 crew members because of the presentation of fever among the crew in April 2020. This was a valuable opportunity to investigate the genetic relationships of SARS-CoV-2 strains transmitted in a large-scale cruise ship cluster.

To analyze phylogenetic relationships and the evolution process of SARS-CoV-2 detected in the CA cruise ship cluster, we performed whole-genome sequencing of CA strains, and also collected sequences of DP strains for the comparison of the evolution. Moreover, to compare the evolution process in cruise ships with that in another environment, we also determined complete genome sequences of SARS-CoV-2 strains isolated from clinical specimens in Japan (FJ Japan strains). Phylogenetic and haplotype network analysis of CA and DP strains revealed the evolution process of the virus in cruise ships as similar to that in a superspreading event. Interestingly, there were significant genetic distances between the populations of cruise ship strains and other strains in different environmental settings. Thus, these results provide evidence of the unique evolution of SARS-CoV-2 in the CA cruise ship cluster.

## 2. Materials and Methods

### 2.1. Ethics Statement

This study was approved by the Institutional Review Board of Nagasaki University (approval no. 200409234).

### 2.2. Clinical Samples

We received nasopharyngeal swab samples for COVID-19 diagnosis between 20–25 April 2020. The tested and COVID-19 positive sample numbers are shown in Appendix A.

### 2.3. Diagnostic Tests

Viral RNA was extracted using a QIAamp Viral RNA Mini Kit (Qiagen, Hilden, Germany) according to the manufacturer’s instructions. We first used a reverse transcription loop-mediated isothermal amplification (RT-LAMP) assay for COVID-19 diagnosis as described previously [19]. Briefly, RT-LAMP was performed using Isothermal Master Mix reagent ISO-004 (Canon Medical Systems, Tochigi, Japan) on a Genelyzer FIII device (Canon Medical Systems). The reaction mixture (total volume: 25 μL) contained 15 μL of Isothermal Master Mix, 1 U of AMV reverse transcriptase (Nippon Gene, Tokyo, Japan), 20 pmol (each) of FIP and BIP primers, 5 pmol (each) of F3 and B3 outer primers, 10 pmol (each) of F and B loop primers, and 5 μL of RNA sample. The reaction was performed under the following conditions: 68 °C for 20 min, followed by a dissociation analysis at 95 °C–75 °C with a temperature change rate of 0.1 °C/s. Synthesized RNAs of the target sequence were used as positive controls. Nonspecific amplification was excluded by comparing the melting temperature with that of the positive control.

When we found COVID-19-positive samples, we confirmed the results by reverse transcription-quantitative PCR (RT-qPCR) using One Step PrimeScript III RT-qPCR Mix (Takara Bio, Shiga, Japan) as reported previously [19]. The reaction mixture (total volume: 20 μL) contained 10 μL of 2× One Step PrimeScript III RT-qPCR Mix, 0.4 μL of ROX Reference Dye, 2 μL of 10× primer-probe mixture, 2 μL of RNA sample, and 5.6 μL of RNase-free water. The primers and a probe were used with final concentrations of 500 nM, 700 nM, and 200 nM of the forward primer, reverse primer, and FAM-labeled probe, respectively. The RT-qPCR reaction was performed using the StepOnePlus instrument (Thermo Fisher Scientific, Waltham, MA, USA) with a thermal cycle program of 52 °C for 5 min and 95 °C for 10 s, followed by 45 cycles of 95 °C for 5 s and 60 °C for 35 s. The cut-off values were set at a threshold cycle (Ct) value of 40. A standard curve was generated using 10-fold serial dilutions of synthesized standard RNA of the target sequence for viral RNA quantification.

### 2.4. Whole-Genome Sequencing of SARS-CoV-2

COVID-19-positive samples with a high viral titer (Ct values < 30 by RT-qPCR) were used for whole-genome sequencing using a method similar to that reported previously [20]. Briefly, multiplex PCR was performed using primer sets designed by the online program Primal Scheme (https://primalscheme.com/ (accessed on 7 April 2020)), with an amplicon size of 450 bp. The template sequence was derived from the Wuhan-Hu-1 strain (GISAID accession no. EPI_ISL_402125). Extracted viral RNA was reverse-transcribed using SuperScript IV Reverse Transcriptase (Thermo Fisher Scientific) combined with random hexamers according to the manufacturer’s protocol. Reverse-transcribed complementary DNA was used to perform multiplex PCR with Q5 High-Fidelity DNA Polymerase (New England Biolabs, Ipswich, MA, USA) as described previously [21]. Libraries were prepared using 40–500 ng of multiplex PCR products and an NEBNext Ultra II DNA Library Prep kit (New England Biolabs) in combination with NEBNext Multiplex Oligos for Illumina (New England Biolabs) according to the manufacturer’s instructions. After a quantitative inspection of each library using an NEBNext Library Quant Kit for Illumina (New England Biolabs), we used the large-scale sequencing service GeneNex (Chemical-dojin, Tokyo, Japan) to obtain complete genome sequences of SARS-CoV-2. Mapping of the paired-end reads was performed on CLC Genomics Workbench 11.0.1 software (Qiagen) using a whole-genome sequence of the Wuhan-Hu-1 strain as a template. Consensus sequences were extracted and aligned with reference strains using BioEdit 7.0.5.3 software (http://www.mbio.ncsu.edu/BioEdit/bioedit.html (accessed on 4 July 2021)). Sanger sequencing was performed to obtain complete genome sequences of several strains that showed short ambiguous sequences.

### 2.5. Phylogenetic Analysis

To infer the phylogeny of SARS-CoV-2 complete genomes, high-coverage reference sequences that possessed more than 29,700 nt of submitted genome length were obtained from GISAID as described previously [22]. Reference sequences were reported worldwide from January to September 2020. Consensus sequences of whole genomes of SARS-CoV-2 strains detected in this study and reference sequences were aligned using the MAFFT web server (https://mafft.cbrc.jp/alignment/server/ (accessed on 4 July 2021)) and checked manually for gaps to be removed. To remove ambiguous nucleotides, 85 nt from the 5′-terminus and 217 nt from the 3′-terminus were excluded from the analysis. Finally, 1985 sequences were included in the analysis. Maximum-likelihood analysis was performed using IQ-TREE software (http://www.iqtree.org/ (accessed on 21 July 2021)) under the condition of a best-fit substitution model GTR+F+R2, including the Wuhan-Hu-1 strain as a root. Although a total of 1000 bootstrap replicates were generated for the analysis, due to the size of the alignment and low variability, it was opted to not infer support for splits in this tree topology [23]. We checked that tree topology on major splits was maintained in another cycle of analysis. For better visualization, the phylogenetic tree was modified using iTOL v6 (https://itol.embl.de/ (accessed on 21 July 2021)). For the phylogenetic analysis of SARS-CoV-2 genome sequences detected in this study with reference strains reported only from Japan, high-coverage reference sequences were retrieved from the GISAID database, and 394 sequences were included in the analysis. Maximum-likelihood analysis was performed using IQ-TREE with a total of 1000 replicates. For better visualization, the phylogenetic tree was modified using FigTree v1.4.2 software (http://tree.bio.ed.ac.uk/softw are/figtree (accessed on 21 July 2021)).

### 2.6. Mutation Site Identification

Complete SARS-CoV-2 sequences of each sample group were aligned using the MAFFT web server with the Wuhan-Hu-1 sequence as a reference. Alignment files were analyzed by the online COVID-19 genome annotator program (http://giorgilab.unibo.it/coronannotator/ (accessed on 2 August 2021)). Single nucleotide variant (SNV) frequency, transition/transversion mutation numbers, and non-synonymous mutation numbers were exported and visualized using GraphPad Prism 7 software (GraphPad Software, San Diego, CA, USA).

### 2.7. Haplotype Network Construction

Aligned complete genome sequences of each sample group (FJ, CA, and DP) were imported into MEGA 7 software to generate files in the nexus format for downstream analysis (https://www.megasoftware.net/ (accessed on 12 August 2021)). Population structure analyses were performed using the nexus files on PopART version 1.7 (http://popart.otago.ac.nz/index.shtml (accessed on 12 August 2021)) with default parameters to generate median-joining haplotype networks. The haplotype networks were modified using PopART for better visualization. The numbers of segregating sites and parsimony-informative sites were also checked using PopART. Haplotype diversity was calculated using DnaSP v5.10.01 (http://www.ub.edu/dnasp/ (accessed on 12 August 2021)).

### 2.8. Pairwise Distance Calculation and Genetic Diversity Analysis

SARS-CoV-2 sequences from lockdown cities (Paris-LD, Milan-LD, and Moscow-LD), a superspreading event (Austria-SSE), and cities/countries in a normal state (England, California, and France) were collected from GISAID and aligned using BioEdit. For SARS-CoV-2 strains circulating in lockdown cities, we selected Paris, Milan, and Moscow for the following reasons: (1) the lockdown period was temporally adjacent to the emergence of the cruise ship CA cluster, and (2) appropriate numbers of sequences were available to compare with CA strains [24]. From Paris, sequences of 236 strains collected between 17 March and 10 May 2020, were retrieved from GISAID. Similarly, 226 and 184 sequences were retrieved from GISAID for Milan from 3 May–11 March 2020, and for Moscow from 12 May–30 March 2020, respectively. The SARS-CoV-2 sequences of a large-scale superspreading event that occurred in Austria included 90 sequences and were aligned for further analyses [25]. For the periods outside lockdown, 426 SARS-CoV-2 sequences from England from 1 January–7 March 2020; 776 from California from 16 April–8 May 2020; and 818 from France from 1 July–31 August 2020, were downloaded from GISAID. The level of genetic differentiation among SARS-CoV-2 strains in each sample group (FJ, CA, DP, Paris-LD, Milan-LD, Moscow-LD, Austria-SSE, England, California, and France) was estimated by generating Fst values in Arlequin v3.5.2.2 (http://cmpg.unibe.ch/software/arlequin35/ (accessed on 12 August 2021)). Neutrality tests, including Fu’s Fs test, were also performed using Arlequin v3.5.2.2.

## 3. Results

### 3.1. Phylogenetic Analysis of SARS-CoV-2 Detected in a Cruise Ship Cluster

Between 20–25 April 2020, 625 pharyngeal specimens were diagnostically tested for COVID-19 (Appendix A). Specimens were derived from crew members of a large cruise ship, Costa Atlantica, which had been in repair since January 2020 in Nagasaki, Japan. The RT-LAMP assay that was developed by the study group was employed for COVID-19 diagnosis, and 148 samples were positive (positivity rate: 23.8%). As the positivity rate for the cluster on another cruise ship, the Diamond Princess, was 19.2%, the CA cluster showed a more serious situation in a highly isolated environment [26] (Appendix A). All positive samples that showed high viral titers were subjected to whole-genome sequencing, and 94 complete genome sequences were obtained for the CA cluster. We also performed whole-genome sequencing using 82 SARS-CoV-2 isolates originating from clinical specimens mainly collected in Tokyo, Japan (FJ Japan strains), to compare evolutionary changes in SARS-CoV-2 genomes with cruise ship strains.

To illustrate the global phylogeny of SARS-CoV-2 sequenced in this study, phylogenetic relationships were inferred with reference strains isolated worldwide (Figure 1). CA strains formed one large cluster without any other reference strains, indicating that CA strains originated from one ancestral strain introduced on the cruise ship and circulated in a highly isolated environment. To infer a potential origin of SARS-CoV-2 introduced into the CA cruise ship, whole-genome sequences of two CA strains (CA071 and CA518) that were located at the root of the CA phylogenetic cluster were subjected to BLAST analyses on GISAID database. Focusing on the SARS-CoV-2 strains detected before sample collection in this study, the result of the BLAST analysis clearly showed that both two CA strains were highly close to a Russian strain hCoV-19/Russia/SPE-57701/2020 (GISAID accession no. EPI_ISL_6565013), strongly indicating a potential introduction of SARS-CoV-2 into the CA cruise ship from European countries. SARS-CoV-2 strains isolated from the DP cluster formed one large phylogenetic cluster, and several DP strains were separated from this cluster, as shown in the previously reported phylogeny of DP strains [17]. Especially, one DP strain shown by the left green arrow in Figure 1 (DP0457) was located at the position genetically far from the DP cluster (Figure 1). To examine whether the origins of these DP strains were distinct from each other, BLAST analysis was performed using the sequences of the DP0078 (located in the DP phylogenetic cluster) and DP0457 (left green arrow in Figure 1) strains. The result showed that the sequence of the DP0078 strain was identical to several strains such as hCoV-19/Japan/TKYE6938/2020 (GISAID accession no. EPI_ISL_468724). Interestingly, the BLAST analysis on the DP0457 strain found the same sequence set as the result on DP0078, although the DP0457 genome possessed three mutations against each BLAST-hit sequence. All CA strains possessed the D614G amino acid substitution in the S protein, whilst all DP strains showed amino acid 614D. FJ strains were widely distributed in the tree, reflecting the various origins of SARS-CoV-2 circulating in Japan. FJ strains contained one original East Asian-type strain (FJ294), and all other FJ strains possessed a D614G substitution in the S protein. Phylogeny inference using reference strains reported only from Japan showed similar genetic characteristics as those of CA and FJ strains (Figure 1), whereas several reference strains were located in the DP cluster (Appendix A).

### 3.2. Characterization of Mutation Sites and Substitution Patterns in CA Strains

To identify SNVs and amino acid substitution patterns specific to each cruise ship cluster, the genome sequences of the FJ, CA, and DP strains were compared with the reference Wuhan-Hu-1 genome sequence. The results showed more SNVs in FJ and CA than in DP strains: FJ, 1191 SNVs and 612 amino acid substitutions; CA, 1175 and 755; DP, 149 and 110 (Table 1). Regarding amino acid substitution numbers, FJ and CA strains possessed 5.56- and 6.86-fold more substitutions per strain than DP strains did, indicating a significant small number of amino acid substitutions in DP strains (Table 1). SNVs tended to be widely distributed in the genome rather than confined to specific genes, although amino acid substitutions observed in more than two strains were found the most in the NSP3 protein in both CA and DP groups (Figure 2a and Table 2). Focusing on the S gene, a variety of non-synonymous mutations were developed in each cluster, although none of the amino acid substitutions was common between the CA and DP cruise ship clusters (Table 3). We further looked for common amino acid substitutions that were acquired in the CA and DP clusters throughout the genome but found no common evolutionary substitutions between the cruise ship clusters, possibly suggesting no cruise ship-specific selective pressures for the virus spread (Appendix A). Comparison of nucleotide substitutions clarified that C > T and G > A transitions were the main mutations in the FJ and CA strains (Figure 2b). Transitions constituted 82.5% of mutations in FJ strains, 67.9% in CA, and 40.3% in DP. CA and DP strains showed cruise ship-specific characteristics in nucleotide substitutions: CA strains possessed a relatively high rate of C > A transversion, and G > T transversion was the most abundant mutation in DP strains (Figure 2b). More non-synonymous mutations existed in all sample groups than synonymous or extragenic mutations, as reported previously (Figure 2c) [17]. The number of amino acid substitutions per strain was larger in CA strains than that in FJ strains, although the number of total mutation sites per strain was smaller in the CA population than that in FJ, indicating more mutations causing amino acid substitutions were selected in the CA cruise ship (Table 1 and Table 3).

### 3.3. Haplotype Diversity in SARS-CoV-2 Strains Detected in Cruise Ship Clusters

To investigate the genetic diversity of SARS-CoV-2 genomes in cruise ship clusters, haplotype networks were constructed using the complete genome sequences of each sample group. The CA strains showed three main clusters, and the CA-A cluster was genetically closest to the reference strain, Wuhan-Hu-1, possibly indicating that the CA-A cluster was the haplotype originally introduced into the CA cluster (Figure 3a). SARS-CoV-2 obtained novel haplotypes on the CA cruise ship to form the CA-B (G25437T, corresponding to L15F in the ORF3a protein) and CA-C clusters (G25437T+C23604A, corresponding to L15F+P681H in the ORF3a and S protein, respectively), finally obtaining a reversion mutation, T11195C, that was originally found in the Wuhan-Hu-1 strain (Figure 3a). The first cluster CA-A was not a large cluster and only two infected individuals possibly played a central role in spreading SARS-CoV-2 in the CA-A cluster. Meanwhile, the core populations of clusters CA-B and CA-C included more than 10 individuals, indicating that these clusters were caused by superspreading event-like infections (Figure 3a). DP strains also showed three main clusters, and the DP-A cluster seemed to be caused by a superspreading event on the cruise ship, including 29 infected individuals with the identical strain to form a large core in the DP-A cluster (Figure 3b). DP strains obtained mutations that resulted in the novel haplotypes DP-B (C18656T, corresponding to T206I in the NSP14 protein) and DP-C (C18656T+C29635T, C29635T was a synonymous mutation) on the cruise ship. In contrast, the FJ strains formed a more complicated network than the cruise ship strains did (Figure 3c). There were five base differences between the FJ-A and FJ-B clusters, indicating that the origins of these clusters were genetically distant from each other (Figure 3c). Moreover, novel haplotypes were actively generated within each FJ cluster, probably reflecting the circulation of SARS-CoV-2 strains of different origins in Japan (Figure 3c). Comparison of segregating and parsimony-informative sites in haplotype networks showed higher complexity in the FJ haplotypes than that in cruise ship clusters, possibly reflecting the existence of multiple origins of SARS-CoV-2 circulating in Japan (Appendix A).

### 3.4. Pairwise Genetic Distance between Cruise Ship Strains and Isolates from Lockdown Cities or Superspreading Events

SARS-CoV-2 transmission on a cruise ship may recapitulate the COVID-19 situation in lockdown cities or large-scale superspreading events owing to their similarities in environment and isolated or crowded places. To analyze the genetic distance between cruise ship clusters and other situations of COVID-19, pairwise distance (Fst) was calculated using SARS-CoV-2 genome sequences detected in lockdown cities (Paris-LD, Milan-LD, and Moscow-LD), a superspreading event in Austria (Austria-SSE), and countries/cities without lockdown (England, France, and California) [24,25]. All Fst values were statistically significant (*p* < 0.001). The results showed remarkable genetic closeness (Fst < 0.1) between lockdown cities (Milan-LD and Moscow-LD) or within the same country (France and Paris-LD), as well as cities without lockdown (California and France), indicating that the analysis using a population genetics method was a reasonable approach for examining evolution of SARS-CoV-2 populations (Figure 4). Cruise ship populations (CA and DP) were significantly differentiated from SARS-CoV-2 circulating in lockdown cities or superspreading events (Fst > 0.2). Interestingly, the population of the CA cruise strains showed a large genetic distance from that of the DP cruise cluster (Fst > 0.7) (Figure 4). Moreover, the Austria-SSE population showed a significant genetic distance from cruise ship populations (Fst > 0.7), although the haplotype network of DP strains showed similar characteristics as those of the Austria-SSE population in segregating and parsimony-informative sites (Appendix A). These results suggest the unique evolution process of the population of cruise ship strains. The SARS-CoV-2 populations of countries/cities with or without lockdown showed a relatively close genetic relationships (Fst < 0.2) except the value between Paris-LD and Moscow-LD, indicating that the environment for the virus spread was similar in such large-scale populations (i.e., countries/cities), and that the cruise ship clusters and a superspreading event (Austria-SSE) were in a unique condition considerably distinct from large-scale populations. FJ strains, detected in Japan without lockdown, showed relatively larger Fst values against the foreign populations without lockdown (Fst: 0.212–0.297) than those between cities without lockdown except Japan (Fst: 0.097–0.171), possibly reflecting the fact that the environmental condition of the Japanese island is dissimilar to other reference regions and promoted the development of Japan-specific haplotypes (Figure 4).

Further analyses using Fu’s Fs test, which is particularly sensitive to population expansion, were performed to investigate the neutrality in evolution of each SARS-CoV-2 population. All values were statistically significant (*p* < 0.01). The results suggested that all populations had experienced expansion. Remarkably, the expansion of genetic diversity in SARS-CoV-2 populations from cruise ships (CA and DP) and a superspreading event (Austria-SSE) was significantly smaller than that of the other SARS-CoV-2 populations from countries/cities with or without lockdown (FJ, England, California, France, Paris-LD, Milan-LD, and Moscow-LD). These results suggest the limited capacity of population expansion in a confined area compared with large-scale populations (countries/cities) (Table 4).

## 4. Discussion

A vast number of surveillance studies have clarified natural mutations in the genomes of circulating SARS-CoV-2 strains, showing the genetic characteristics of the virus in each country/region. Although the viral genome is relatively stable because of the proofreading function of viral RNA polymerase, several critical mutations have provided SARS-CoV-2 with enhanced transmissibility and an immune-escape ability [5,6,7,8]. Therefore, the investigation of the evolutionary process of SARS-CoV-2 is essential to understand the virus-specific genetic characteristics and prepare for the emergence of novel variants.

The present study investigated the evolution of SARS-CoV-2 strains in the second large cruise ship cluster in Japan on the Costa Atlantica (CA). Recently, an epidemiological study revealed the relationships of SARS-CoV-2 infections with age, occupation, underlying diseases, symptoms, and cabin room location on the CA cruise ship [18]. In this previous study, the first case of a body temperature >37.1 °C was identified on 22 March 2020, in a crew member who belonged to the entertainment group that boarded the CA cruise ship from Europe on March 18 or 19 [18]. Our BLAST analysis using the sequences of CA strains located at the root in the phylogenetic tree revealed that the possible origin of CA strains was derived from European countries, showing a high consistency of our results with the previous study. Assuming that SARS-CoV-2 was first introduced on 18 March, the evolutionary rate of SARS-CoV-2 on the CA cruise ship was estimated to be approximately 2.80 × 10^−3^ substitutions per site per year. However, a number of past studies have estimated the evolutionary rate of SARS-CoV-2 to be 0.99–1.69 × 10^−3^ substitutions per site per year [27,28,29,30]. The estimation of a higher evolutionary rate on the CA cruise ship indicates that the environmental conditions of the CA cluster were unique compared with those of other clusters.

The most well-known and studied cruise ship cluster is the DP cluster, which emerged in Japan in February 2020. A number of epidemiological and clinical analyses of the DP cluster have provided important evidence regarding the contribution of asymptomatic patients and estimation of the reproductive number of the cluster [13,14,15,16]. Compared with those epidemiological studies, only one report of genetic analyses of SARS-CoV-2 in the DP cluster has been published, and the study well described the transmission process as similar to that in a superspreading event [17]. Our results provide evidence of a similar transmission process in the CA cluster (Figure 3). However, the number of mutations in the DP cluster was remarkably lower than that in the CA cluster, indicating environmental differences between the two cruise ships. The BLAST analysis of several DP strains, even in the DP strain separated from the phylogenetic DP cluster, found the same sequence set as a potential origin. This result indicates the single introduction of SARS-CoV-2 into the DP cruise ship, and thus there would be no difference in SARS-CoV-2 introduction between the CA and DP cruise ships. Of all 712 COVID-19 positive individuals, 17.9% were estimated to be asymptomatic in the DP cluster [13]. Interestingly, 35.6% of COVID-19 positive cases were asymptomatic in the CA cluster, indicating that more asymptomatic individuals were related with the spread of the virus in the CA cruise ship than in DP [18]. Moreover, a recent study provided evidence that at least 50% of SARS-CoV-2 transmission is caused by asymptomatic individuals [31]. These reports suggest that SARS-CoV-2 circulation on the CA cruise ship might have begun before March 22 as asymptomatic cases, and that more asymptomatic infection cases had been unrecognized in the CA cruise ship, generating much more genetic diversity compared with that in the DP cluster.

Around half of the CA strains acquired the P681H substitution, one of the main mutations in the B.1.1.7 lineage, in the S protein (Figure 2 and Figure 3, Table 3). B.1.1.7 strains show higher transmissibility and have been defined as a variant of concern [6,32,33]. Although the most important substitution in the lineage is considered to be N501Y in the S protein, P681H is adjacent to the furin cleavage site (A684/R685) and may affect the efficiency of cleavage, which facilitates efficient SARS-CoV-2 transmission and infection [34]. Therefore, P681H may be the key substitution that enhanced the transmissibility of SARS-CoV-2 on the CA cruise ship; however, whether P681H affects the cleavage of the S protein by furin protease requires further investigation. Furthermore, we found a relatively high rate of SNVs on the NSP3 protein in cruise ship strains (Table 2). Previous studies demonstrated that the conserved macrodomain encoded in the NSP3 protein of coronaviruses was essential for viral replication and pathogenicity in the animal host [35,36]. Critical mutations in the macrodomain of the NSP3 protein resulted in the attenuation of coronavirus pathogenicity due to enhanced innate immune responses [37]. Thus, the NSP3 protein plays essential role in the coronavirus spread. The increased number of NSP3 mutations may explain the unique environment for virus transmission in cruise ships.

A very small number of large cruise ship clusters have been reported [25,38]. In the US, the Grand Princess cruise ship experienced a large cluster in March 2020; 78 individuals were confirmed to be COVID-19 positive (positivity rate 16.6%). In Australia, approximately 2700 passengers disembarked from the Ruby Princess cruise ship at the port of Sydney in March 2020; among them, 162 individuals were COVID-19 positive [39]. These cases would have been valuable for analyzing SARS-CoV-2 evolution and genome diversity acquired in crowded and confined places. However, reports of genetic analyses of SARS-CoV-2 in cruise ship clusters are scarce. In addition, information on the temporal pattern of evolution is unavailable using the clinical diagnostic samples collected during a very short sampling period. Further genetic analyses using clinical specimens derived from cruise ship clusters will be informative for understanding the SARS-CoV-2 transmission process in isolated places.

The main limitations of the study include unavailability of detailed epidemiological data to identify key environmental factors enhancing mutations in the CA strains. The demographic information of crew members associated with each SARS-CoV-2 whole-genome sequence might have been informative to analyze environmental factors for the unique evolution in the CA cruise ship. However, it is always difficult to collect epidemiological information during the urgent COVID-19 diagnostic tests. An efficient diagnosis system should be explored to collect epidemiological data available even when an urgent diagnostic test in collaboration with clinicians and public administration. The research using both genomic and epidemiological data will be highly informative for the development of a countermeasure to efficiently prevent the virus spread in similar settings.

The present study provides new insights into the evolution of SARS-CoV-2 in an isolated and confined place. Recently, novel variants have emerged worldwide. Further field epidemiological studies are required to identify the original cause of infections, together with genetic analyses of SARS-CoV-2, to monitor and properly prepare for novel mutants emerging in specific environments.

## Figures and Tables

**Figure 1 microorganisms-10-00099-f001:**
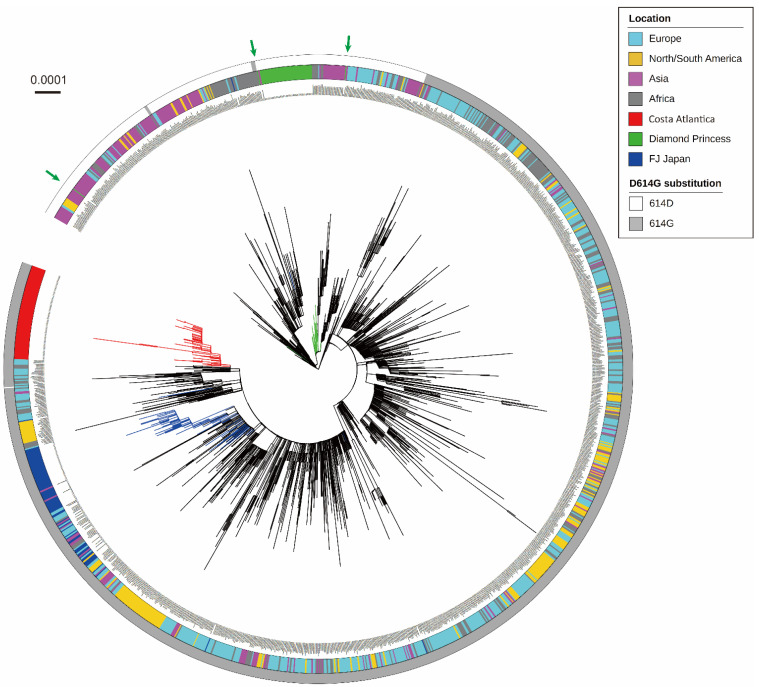
The maximum likelihood phylogeny of SARS-CoV-2 whole-genome sequences detected in the Costa Atlantica and Diamond Princess cruise ship clusters, together with the reference sequences worldwide, including the Wuhan-Hu-1 strain (GISAID accession no. EPI_ISL_402125). Branches of the three SARS-CoV-2 populations (Costa Atlantica, Diamond Princess, FJ Japan) are colored in red, green, and blue, respectively. The multicolored ring shows the location of sample collection for each strain. The presence (gray) or absence (white) of the D614G substitution is indicated in the outer ring. Green arrows show the strains from the Diamond Princess that were separated from the Diamond Princess cluster. The scale bar depicts the nucleotide substitutions per site.

**Figure 2 microorganisms-10-00099-f002:**
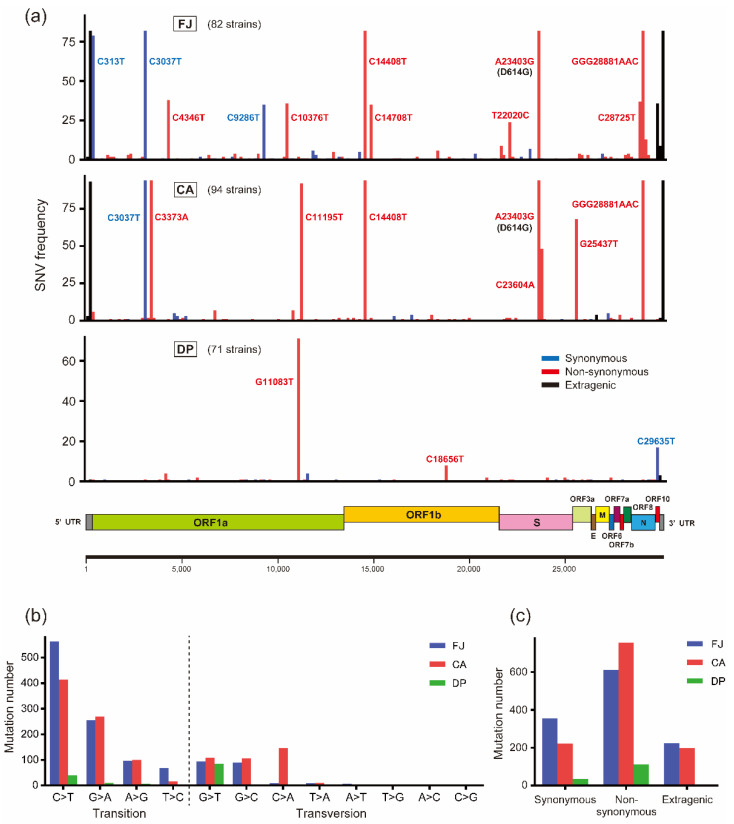
Single nucleotide variants of SARS-CoV-2 strains on cruise ship clusters. (**a**) The genome position of nucleotide variations found among the FJ Japan, Costa Atlantica, and Diamond Princess strains compared with the Wuhan-Hu-1 strain (GISAID accession no. EPI_ISL_402125). Nucleotide positions are indicated for the main variants. Synonymous, non-synonymous, and extragenic variants are depicted in blue, red, and black, respectively. The genome of SARS-CoV-2 is displayed under the SNV frequency graph. SNV, single nucleotide variant; FJ, FJ Japan; CA, Costa Atlantica; DP, Diamond Princess. (**b**) The number of detected nucleotide transitions and transversions in FJ, CA, and DP strains relative to the Wuhan-Hu-1 strain. (**c**) The number of detected synonymous, non-synonymous, and extragenic mutations on the genome sequence of SARS-CoV-2 relative to the Wuhan-Hu-1 strain.

**Figure 3 microorganisms-10-00099-f003:**
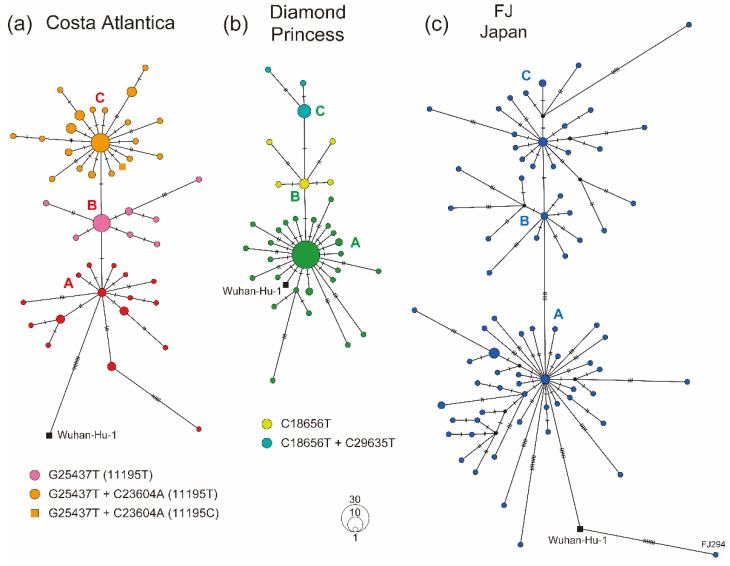
Haplotype network using whole-genome sequences of the cruise ship cluster strains. (**a**) Haplotype of the Costa Atlantica strains inferred using a median-joining single nucleotide variant network analysis. The presumed first spread cluster (cluster A) is indicated in red, and G25437T and G25437T+C23604A variants (clusters B and C) are in magenta and orange, respectively. (**b**) Haplotype of the Diamond Princess strains. The presumed first spread cluster (cluster A) is indicated in green, and C18656T and C18656T+C29635T variants (clusters B and C) are in light green and light blue, respectively. (**c**) Haplotype of the FJ Japan strains. Small tick marks on branches depict the number of single nucleotide variants between haplotypes.

**Figure 4 microorganisms-10-00099-f004:**
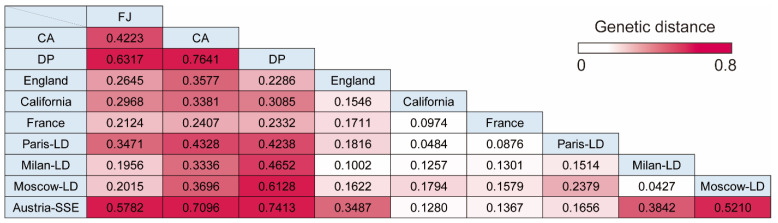
Pairwise distances among different SARS-CoV-2 groups. Genetic distances are indicated in magenta that is graded according to the distance. FJ, FJ Japan; CA, Costa Atlantica; DP, Diamond Princess; LD, Lockdown; SSE, Superspreading event.

**Table 1 microorganisms-10-00099-t001:** Number of mutations and amino acid substitutions in each SARS-CoV-2 group.

Group	Sample	Strain	Mutations		Amino Acid Substitutions	
			Total Number	Number /Strain	Total Number	Number/Strain
FJ Japan	Clinical tests	82	1191	14.52	612	7.46
Costa Atlantica	Cruise ship	94	1175	12.50	755	8.03
Diamond Princess	Cruise ship	71	149	2.10	110	1.55

**Table 2 microorganisms-10-00099-t002:** Representative amino acid substitutions observed in more than two strains in each SARS-CoV-2 group.

FJ Japan (*n* = 82)						Costa Atlantica (*n* = 94)					
Position ^1^	Ref ^2^	Mut ^3^	Protein	Substitution	*n*	Position	Ref	Mut	Protein	Substitution	*n*
14408	C	T	ORF1b NSP12b	P314L	82	3373	C	A	ORF1a NSP3	D218E	94
23403	A	G	S	D614G	82	14408	C	T	ORF1b NSP12b	P314L	94
28881	GGG	AAC	N	RG203KR	82	23403	A	G	S	D614G	94
4346	T	C	ORF1a NSP3	S543P	35	28881	GGG	AAC	N	RG203KR	94
10376	C	T	ORF1a NSP5	P108S	35	11195	C	T	ORF1a NSP6	L75F	92
14708	C	T	ORF1b NSP12b	A414V	35	25437	G	T	ORF3a	L15F	64
28725	C	T	N	P151L	35	23604	C	A	S	P681H	44
22020	T	C	S	M153T	23	10755	C	T	ORF1a NSP5	A234V	7
28975	G	T	N	M234I	13	376	G	C	ORF1a NSP1	E37D	6
21518	G	T	ORF1b NSP16	R287I	7	6660	G	A	ORF1a NSP3	S1314N	5
18167	C	T	ORF1b NSP14	P43L	6	17876	C	T	ORF1b NSP13	T547I	4
7728	C	T	ORF1a NSP3	S1670F	4	25445	G	T	ORF3a	G18V	4
12869	A	T	ORF1a NSP9	T62S	3	2901	T	C	ORF1a NSP3	V61A	2
2167	G	T	ORF1a NSP2	K454N	2	3230	G	T	ORF1a NSP3	G171C	2
2910	C	T	ORF1a NSP3	T64I	2	3514	G	T	ORF1a NSP3	M265I	2
4309	G	T	ORF1a NSP3	K530N	2	5018	G	T	ORF1a NSP3	D767Y	2
11083	G	T	ORF1a NSP6	L37F	2	13080	T	C	ORF1a NSP10	F19S	2
21575	C	T	S	L5F	2	13624	G	T	ORF1b NSP12b	D53Y	2
21614	C	T	S	L18F	2	13922	A	G	ORF1b NSP12b	D152G	2
23481	C	T	S	S640F	2	19885	AA	CG	ORF1b NSP15	K89R	2
26966	T	A	M	H148Q	2	27688	C	T	ORF7a	P99S	2
27925	C	A	ORF8	T11K	2	28302	G	T	N	R10L	2
**Diamond Princess (*n* = 71)**											
**Position**	**Ref**	**Mut**	**Protein**	**Substitution**	** *n* **						
11083	G	T	ORF1a NSP6	L37F	71						
18656	C	T	ORF1b NSP14	T206I	8						
4127	G	A	ORF1a NSP3	G470S	2						
5845	A	T	ORF1a NSP3	K1042N	2						

^1^ Nucleotide position in the Wuhan-Hu-1 strain. ^2^ Nucleotide in the reference strain (Wuhan-Hu-1). ^3^ Nucleotide mutations identified in each group.

**Table 3 microorganisms-10-00099-t003:** Amino acid substitutions in the spike gene.

FJ Japan (*n* = 82)					Costa Atlantica (*n* = 94)					Diamond Princess (*n* = 71)				
Pos ^1^	Ref ^2^	Mut ^3^	Sub ^4^	*n*	Pos	Ref	Mut	Sub	*n*	Pos	Ref	Mut	Sub	*n*
21575	C	T	L5F	2	21721	C	A	D53E	1	21575	C	T	L5F	1
21614	C	T	L18F	2	21765	T	G	I68R	1	21917	A	G	I119V	1
21707	C	T	H49Y	1	22021	G	T	M153I	1	22104	G	T	G181V	1
22020	T	C	M153T	23	22275	T	A	F238Y	1	23856	G	T	R765L	1
22199	G	C	V213L	1	22289	G	T	A243S	1	24797	C	A	P1079T	1
22317	G	T	G252V	1	23403	A	G	D614G	94	24819	A	G	K1086R	1
23403	A	G	D614G	82	23604	C	A	P681H	44	25244	G	T	V1228L	1
23481	C	T	S640F	2	23705	C	A	P715T	1					
24328	G	C	L922F	1										
25088	G	T	V1176F	1										
25317	C	A	S1252Y	1										

^1^ Nucleotide position in the Wuhan-Hu-1 strain. ^2^ Nucleotide in the reference strain (Wuhan-Hu-1). ^3^ Nucleotide mutations identified in each group. ^4^ Amino acid substitutions in each group.

**Table 4 microorganisms-10-00099-t004:** Neutrality tests of SARS-CoV-2 populations detected in different settings.

Group	FJ	CA	DP	England	California	France	Paris-LD	Milan-LD	Moscow-LD	Austria-SSE
Fu’s Fs	−24.7827	−10.2815	−9.5685	−23.9885	−23.7186	−23.4760	−24.4464	−24.5018	−25.1710	−13.5053

FJ, FJ Japan; CA, Costa Atlantica; DP, Diamond Princess; LD, Lockdown; SSE, Superspreading event.

## Data Availability

Genomic data of newly sequenced samples were deposited in the GISAID database with accession numbers EPI_ISL_4418186–4418360 and EPI_ISL_4470095–4470096.

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
