# Peer review of "Unique Evolution of SARS-CoV-2 in the Second Large Cruise Ship Cluster in Japan"

_microorganisms, 2022, doi:10.3390/microorganisms10010099_

Round 1

Reviewer 1 Report

The manuscript describes an in depth comparative study of the SARSCoV2 sequence diversity which emerged in a cruise ship outbreak. The methods chosen are adequate and the interpretation of the data is easy to follow.  I found the  description of the data in Table 4 in the results section hard to follow and would suggest to use colour to highlight some of the discussed values. AIternatively I would encourage the authors to try a PCA analysis of the data set which might be helpful in visualising the data groups described.

Author Response

We appreciate the reviewer’s important suggestion to our manuscript. According to the suggestion, we attempted a PCA analysis using sequences of each SARS-CoV-2 group. However, we found that it was difficult to show genetic distances using the result of the PCA analysis, because it does not show accurate genetic distances between each group. Therefore, according to the reviewer’s another suggestion, we put colors in Table 4 and changed it to “Figure 4”. We believe that Figure 4 will help readers to easily understand the genetic distance data.

Reviewer 2 Report

The manuscript entitled "Unique evolution of SARS-CoV-2 in the second large cruise ship cluster in Japan" by Haruka Abe et al provides a unique opportunity to follow the evolution of SARS-CoV-2 within a localized population in a limited time. The manuscript is well written, the data are presented properly and altogether corroborates with other published papers in the field. I find this manuscript suitable for publication in the journal "Microorganisms". 

Good luck

Author Response

We appreciate the reviewer’s kind and significant evaluation of our manuscript.